# Pork Fat and Meat: A Balance between Consumer Expectations and Nutrient Composition of Four Pig Breeds

**DOI:** 10.3390/foods12040690

**Published:** 2023-02-05

**Authors:** Irina Chernukha, Elena Kotenkova, Viktoriya Pchelkina, Nikolay Ilyin, Dmitry Utyanov, Tatyana Kasimova, Aleksandra Surzhik, Lilia Fedulova

**Affiliations:** V. M. Gorbatov Federal Research Center for Food Systems, Experimental Clinic and Research Laboratory for Bioactive Substances of Animal Origin, Moscow 109316, Russia

**Keywords:** pig breed, Livny breed, Altai breed, backfat, consumer preference

## Abstract

Food fat content is one of the most controversial factors from a consumer’s point of view. Aim: (1) The trends in consumer attitudes towards pork and the fat and meat compositions in Duroc and Altai meat breeds and Livny and Mangalitsa meat and fat breeds were studied. (2) Methods: Netnographic studies were used to assess Russian consumer purchasing behavior. Protein, moisture, fat, backfat fatty acid content from pigs, longissimus muscles, and backfat from (A) Altai, (L) Livny, and (M) Russian Mangalitsa breeds were compared with those from (D) Russian Duroc. Raman spectroscopy and histology were applied to the backfat analysis. (3) Results: The attitude of Russian consumers to fatty pork is contradictory: consumers note its high fat content as a negative factor, but the presence of fat and intramuscular fat is welcomed because consumers positively associate them with better taste, tenderness, flavor, and juiciness. The fat of the ‘lean’ D pigs did not show a “healthy” fatty acid ratio, while the n-3 PUFA/n-6 PUFA ratio in the fat of the M pigs was the best, with significant amounts of short-chain fatty acids. The highest UFA content, particularly omega 3 and omega 6 PUFA, was found in the backfat of A pigs with a minimum SFA content. The backfat of L pigs was characterized by a larger size of the adipocytes; the highest monounsaturated and medium chain fatty acid contents and the lowest short-chain fatty acid content; the ratio of omega 3 to omega 6 was 0.07, and the atherogenicity index in L backfat was close to that of D, despite the fact that D pigs are a meat type, while L pigs are a meat and fat type. On the contrary, the thrombogenicity index in L backfat was even lower than the D one. (4) Conclusions: Pork from local breeds can be recommended for functional food production. The requirement to change the promotion strategy for local pork consumption from the position of dietary diversity and health is stated.

## 1. Introduction

Global meat production increased by 5% to 339 million tons in 2021, driven by a significant 34% increase in pork production. The main driver of global meat consumption growth is global population growth, which will continue, and global meat consumption per capita will increase to 35.4 kg by 2027 (by 0.3% each year), an increase of 1.1 kg from 2018 to 2020 [1].

About 30% of all meat consumed in the world is pork. China—the special economic regions of Hong Kong, Macao, and Mainland China—remained the largest pork consumers in 2021, with a consumption of about 61, 52, and 37 kg per capita, respectively. Belarus is the world’s second largest pork consumer, with the European Union, South Korea, and Vietnam ranking third, fourth, and fifth [2].

Pork consumption in Russia in 2021 was 27 kg/person, and this sector grew faster than any other meat group. In comparison, back in 2014, pork consumption was 23–24 kg/year [3].

In addition to population growth, meat consumption is influenced by factors such as income (standard of living), product price levels, demographics (such as an aging population), urbanization rates, traditions, and religious beliefs, as well as public health factors and increasing problems in this area. This is particularly evident now in relation to overcoming the effects of COVID-19.

We analyzed whether this increase in consumption is only due to socio-economic reasons or whether it is also due to some kind of consumer preference for pork.

It is known that one of consumers’ main choice factors with respect to pork is quality. Meat quality is a broad concept that includes organoleptic properties (appearance, tenderness, juiciness, aroma, flavor), nutritional quality, and quality indicators related to food safety. In addition to quality, freshness, price, origin, and fat content are important factors for consumers when choosing pork [4]. It is important to highlight another factor influencing the consumption of pork—the growing trend toward the importance of animal welfare and care for the environment. This trend is gaining momentum, especially among younger generations [5,6], as well as high-income consumers who are switching to a diet that reduces or eliminates the consumption of red meat and meat products and redistributes the food basket towards white/lean meat. In a study [7], changes in consumer preferences were correlated with a change in lifestyle promoted by public health bodies by reducing red meat consumption and moderating fatty pork consumption. However, research shows [8] that the proportion of consumers worldwide that are prepared to stop or significantly reduce their consumption of meat to protect the environment is low. Consumers habitually buy pork; they choose fresh and minced meat, and they most often buy locally produced meat from local butchers.

The key factor in shaping consumer attitudes towards a product is information. The nature of information can be positive, describing benefits, negative, listing risks, or balanced. However, negative information is more persistent than positive information and is related to the instinctive desire of consumers to avoid products for which there is negative information [9,10]. According to [11], the shaping of consumer opinions about a product can be divided into two types: functional and constructive attitudes. While the former remains in a consumer’s memory for a long time, the second group of motivations is created in situ to make a single decision in response to specific circumstances.

There is a discrepancy in consumers’ opinions between what products they want to eat, what they think they eat, and what they actually eat. A clear illustration of this observation is consumers’ attitudes toward pork fat. On the one hand, the fat content in pork may be a factor that can potentially lead to a reduction in the appeal of a product. On the other hand, consumers in many countries do not consider the fat content of pork as a significant factor when choosing a product in a supermarket [12].

The fat content in pork, like any product, is now viewed as a complex of fatty acids that play both positive and negative roles in the prevention of metabolic diseases such as atherosclerosis, myocardial infarction, stroke, obesity, type 2 diabetes, etc. The generally accepted dietary recommendations aim at an overall reduction in saturated fatty acid intake (SFA) and an increase in unsaturated fatty acid intake (UFA). There is a cognitive distortion among the population that pork contains SFA exclusively. However, UFAs account for 42% to 65% of all fatty acids, depending on the breed [13].

There is a preconception that backfat from “lean” meat pigs may be more beneficial for health.

By distinguishing between SFA, MUFA, and PUFA groups, it is advisable to ascertain not only the impact on the physiological processes of each group and their absolute content in different products but also their ratio [14]. The ∑PUFA/∑SFA index is used to show atherogenicity. Its use indirectly assesses the impact of diet on cardiovascular health and is based on the recognized hypothesis that certain unsaturated FA can suppress LDL and contribute to lower blood cholesterol levels, and vice versa with saturated FA. This means that the higher the ∑PUFA/∑SFA index, the greater are the potential health benefits of a product. Additionally, to predict the impact on health, nowadays, attention is paid to the n-3 PUFA/n-6 PUFA ratio, which is skewed towards n-6 in pork, which is not in line with current dietary recommendations. However, the 0.2 ratio recommended by nutritionists needs to be maintained in the overall dietary pattern.

There is a preconception that backfat from “lean” meat pigs may be more beneficial for health. Given that meat fat content is directly related to fatty acid composition, which in turn is closely related to fat deposition and hence to intramuscular fat content and backbone fat thickness [15], it is essential that meat and fat quality characteristics were studied for two meat and two meat and fat pig breeds of Russia.

The aim of the study was to examine trends in consumer attitudes towards pork and assess fat and meat composition in Duroc and Altai meat breeds and Livny and Mangalitsa meat and fat breeds.

## 2. Materials and Methods

### 2.1. Marketing Research

Russian scientific digital library sources (elibrary.ru), portals (https://meat-expert.ru/(accessed on 1 November 2022)), statistical materials (https://rosstat.gov.ru/(accessed on 1 November 2022)) from 2017 to 2022 were used to search for articles. Default terms and the logical connectors “AND” and “OR”: “pork” AND (purchasing behavior OR purchase intent) OR (customer satisfaction) OR (consumer ratings) OR (consumer perception) were used to select studies.

Following questions were asked: (i) How often do you buy pork/minced pork? (ii) How often have you bought chilled pork, frozen pork, et cetera over the past 1–2 years?

The consumers had to answer by choosing options- I buy less now/ or rarely, I buy about the same amount (nothing changed), I start buying more often, I never buy pork minced meat/I never buy pork.

Then the respondents were asked to answer the questions: (1) Why do you buy pork/minced pork; and (2) Why do you not buy pork/minced pork?—by choosing the following options—affordable price/high price, dietary diversity/reasonability (only in warm periods or for cooking special grilled meat for holidays), satisfied/not satisfied with quality (fat content), easy/hard to cook, good/not good sensory characteristics (taste, appearance, juiciness).

The titles of the retrieved articles were analyzed; some publications were excluded (documents whose full texts could not be found were discarded in addition to those defined according to the following criteria: (a) consumer purchasing behavior for “non-pork” meat; (b) documents with a sample size of less than 150–200 participants [16]; (c) documents that do not present specific factors/user assessments).

During the analysis of Russian documents that matched the search characteristics of our study, both for pork and for variables related to purchasing behavior, a total of 107 documents were found in the first stage. Of these, after sifting out duplicates that did not fit the empirical article and matching the exclusion criteria, only 15 studies remained for analysis. At the same time, 3 regions were found in which studies were conducted, which were distributed as follows: Central Federal District (Moscow-4, St. Petersburg-3), Siberian Federal District (Altai Krai-3, Kemerovo Oblast-2), Southern Federal District (Rostov Oblast-2). Based on the data presented in the articles [17,18,19,20,21], the responses were categorized into “rarely” and “often”, shaping the factors that determine consumer assessments.

Netnography was used in this study to assess the attitudes of Russian consumers in the minced meat category. Netnography refers to a qualitative research methodology that adapts ethnographic research methods to the study of consumer behavior [22,23]. Content sampling was conducted for the years 2020 to 2021 and the study covered the Russian Federation. During the study, 9000 consumer reviews evaluating and describing the experience of buying and using minced meat, including more than 270 reviews of minced pork, were collected and processed by machine. A Voice Monitor (a system for machine-based data collection, intelligent data processing and interpretation), which was developed in-house by the V.M. Gorbatov Federal Research Centre for Food was used as an analytical tool for the study. The following indicators were highlighted in the analysis: key consumer choice factors in respect of the product in the minced meat category, a sentiment analysis of consumer attitude towards the product by choice factor, customer satisfaction with the product and experience of use, as well as a semantic analysis of content for different categories of minced meat.

### 2.2. Animals and Sample Preparation

Four pig breeds were used: Livny (*n* = 6; L); Duroc (*n* = 6; D); Altai meat breed (*n* = 5; A); Mangalitsa (*n* = 5; M).

The Livny breed (Livny, Oryol region) is a local Russian breed registered in 1949 (Long-eared white, Yorkshire, Large White, Berkshire boars were used in its breeding), and is a meat and fat breed.

Duroc pigs (Moscow Region) are widely used in the Russian Federation as a commercial, highly productive meat breed.

The Altai breed (Altai Republic) is a local Russian breed registered in 2017 (Large White and Landrace pigs, boars of the Maxgrow terminal line were used in its breeding) and is a meat breed.

The Livny, Duroc, and Altai pigs were kept under the conditions of a commercial pig farm and consumed complete feed.

Mangalitsa pigs (imported from Hungary to Russia in 2000) were reared free range with access to pasture grazing (Krasnoyarsk District).

When the mean live weight of pigs reached 110 ± 10 kg, the pigs were transported to a slaughterhouse, rested for 12 h, and then slaughtered and chilled overnight. They were slaughtered without electrical stimulation and cooled at 0 °C for 24 h in cold storage. The cold carcasses were graded 24 h postmortem. Backfat (BF) in between 10th and 11th rib and longissimus muscle from the left side of the carcass and fixed location between the 5th and 13th rib samples were removed and sent to the laboratory. Pieces of backfat 5 × 5 cm, depth—from subcutaneous fat surface to muscle layer, approximately 70–150 g (±5 g) and l. dorsi, 500 g (±10 g) were cut from each carcass. The samples were obtained from at least three replicates from each carcass; average was used for further data processing.

### 2.3. Proximal Analysis

Methods included Kjeldahl for protein, Soxhlet with acid hydrolysis for fat, drying- and vacuum-oven methods for moisture and ash, and capillary tube method for fat melting point [24] assessment.

### 2.4. Chromatography of Fatty Acids

For the chromatographic analysis, the sample was melted on a laboratory tile. to obtain fatty acid methyl esters 200 µL of melted fat was taken and transferred into a 15 mL centrifuge tube. A total of 2 mL of a 2M potassium hydroxide solution in methanol was added, and then 4 mL of hexane was added and centrifuged for 1–3 min at 3000–5000 rpm. After centrifugation, 200 µL was taken from the upper hexane layer and transferred to a chromatographic vial, and 800 µL of pure hexane was added to dilute the fatty acid concentration. The obtained sample was analyzed by an Agilent 7890 gas chromatograph with a flame ionization detector and Agilent HP 5 30 m × 0.32 mm × 0.25 μm capillary column (carrier gas: nitrogen) (Agilent Technologies, Inc., CA, USA). Fatty acid data is presented as g/100 g total fatty acids. Determination of the fatty acid composition was performed according to the reported method in the literature [25], with the author’s modification.

Such nutritional indices for assessing of fatty acids composition as ΣSFA, ΣUFA, ΣMUFA, ΣPUFA, ΣHUFA, ΣPUFA/ΣSFA, Σn−3 PUFA, Σn−6 PUFA, Σn−3 PUFA/Σn−6 PUFA, C 18:2/C 14, C 18:1/C 14, ΣFAshort (from C4 to C 10), ΣFAmedium (from C11 to C 16), ΣFAlong (>C 17), and ΣC4-C16/ΣC17-C24 were calculated, and the atherogenic index (IA) and thrombogenicity (IT) were calculated according to the following equations [14]:ΣPUFA/ΣSFA = n−6 PUFA/n−3 PUFA(1)
IA = [C12:0 + (4 × C14:0) + C16:0]/ΣUFA(2)
IT = [C14:0 + C16:0 + C18:0]/0.5 × MUFA + 0.5 × PUFA-n6 + 3 × PUFA-n3 + PUFA-n3/PUFA-n6(3)

### 2.5. Histological Analysis

The BF samples obtained were preserved in 10% Neutral Buffered Formalin for 72 h (24 ± 2 °C). A 1.5 × 1.5 × 0.5 cm slice was then taken from each sample, washed with cold running water for 4 h, and thickened in gelatin at an increasing concentration (12.5%, 25%) (AppliChem GMBH, Darmstadt, Germany) at 37 °C for 8 h each using a thermostat (SPU, Russia). At least three 14 µm thick sections from each slice were made on a MIKROM-HM525 cryostat (Thermo Scientific, MA, USA). The sections were placed on Menzel-Glaser slides (Thermo Scientific, MA, USA) and stained with Ehrlich’s hematoxylin and a 1% aqueous-alcoholic eosin solution (BioVitrum, Moscow, Russia) according to the generally accepted technique [26]. The sections were encased in glycerol gelatin. Histological samples were studied using an AxioImaiger A1 light microscope (Carl Zeiss, Oberkochen, Germany) with an AxioVision 4.7.1.0 computer image analysis system (Carl Zeiss, Oberkochen, Germany).

To calculate the adipocyte area from formalin-fixed samples, 14 µm-thick sections were prepared and placed on Menzel-Glaser slides (Thermo Scientific, MA, USA), embedded in a drop of saline, and immediately examined with an image analysis system (modified methodology by Velotto S. [27]. An area of at least 150 adipocytes of the outer (just under the skin) and inner (closer to the muscle) layers was measured interactively with an accuracy of ±1.0 µm^2^.

### 2.6. Raman Spectroscopy

Raman spectroscopy was used to estimate the total fatty acid profile of the samples [28]. Spectra were collected on a Renishaw inVia Reflex confocal Raman dispersive spectrometer (Renishaw plc, Wotton-under-Edge, UK) using a 785 nm laser. The spectrometer was calibrated by recording the Raman spectrum of a silicon crystal wafer at 520 cm^−1^ (exposure time 1 s, laser power 10 mW, 1 scan). An L50× magnification lens was used to focus the laser on the surface of the samples. Applicable laser power 100 mW, exposure time 10 s, 3 scans. The laser power and integration time were carefully optimized to avoid photodegradation of the fat samples. Measurements were recorded over a detection range of 800–1800 cm^−1^. At least 6 spectra were recorded for each sample. The spectral analysis and preprocessing (cosmic ray removal, baseline correction using intelligent polynomial algorithms, smoothing using the Savitzky-Golay algorithm, normalization) were performed using Renishaw WiRE 5.2 software (Renishaw plc, Wotton-under-Edge, UK). To determine the relative content of unsaturated fatty acids in each sample, the intensity ratios of the Raman light scattering signals were calculated corresponding to unsaturated bonds to those of saturated or ester bonds. Nine intensity ratios were used: I_970_/I_1297_, I_970_/I_1430_, I_970_/I_1735_, I_1266_/I_1297_, I_1266_/I_1430_, I_1266_/I_1735_, I_1655_/I_1297_, I_1655_/I_1430_, and I_1655_/I_1735_. Peak area and peak intensity were determined.

### 2.7. Statistical Analysis

The STATISTICA 17.0 software was used in this study for statistical analysis. The results were calculated as mean ± SD. Significant differences were calculated by one-way ANOVA followed by Duncan’s multiple comparisons. Differences with *p*-values < 0.05 were considered statistically significant. The critical value of the Pearson correlation coefficient is 0.5140 at *p* ≤ 0.05 [29].

## 3. Results and Discussion

Nationwide statistics show that in 2018, 70.6% of Russians over 14 years of age eat pork daily or several times a week [30], with an average of over 20.7 kg per Russian [31]. According to preliminary data, in 2021 Russians consume, on average, 34 kg per year, which is 44% of the total consumption [32].

Figure 1A shows the data about motives for choosing pork by Russian consumers collected during the content analysis of studies, that were chosen for analysis. It was found in the documents [17,18,19,20,21] that the main factors influencing consumer behavior include seasonality (increase purchase frequency in summer) and the “attributes” of pork—fat content, taste, and appearance. Factors that shape consumer preference for pork include taste, fast cooking, and reasonable cost. Constraining factors in the choice of pork were the switch to more “healthy types” of meat, cost, and dissatisfaction with quality (Figure 1A).

In other words, Russian consumer purchasing behavior towards pork meat is driven by the same reasons as in other countries. The predominant factors influencing the meat purchasing decision process in Norway are taste, ease of preparation, and price [8]; in Japan, the factors are marbling, an attractive appearance, the absence of negative product information, and local raw materials [33]; in Spain, the consumer is primarily concerned with safety, animal welfare, and the environment, combined with a hedonistic approach to food. Interestingly, 73% of Spanish consumers surveyed preferred animal-friendly production products, but about 70% of consumers are not willing at all or are willing to pay only 5% more for such products. At the same time, 88% were in favor of products made from local raw materials [34]. Australian consumers ranked taste, juiciness, and odorlessness as the three most important factors, according to 15 intrinsic and 31 extrinsic criteria in relation to pork meat, and “animal welfare”, “country of origin”, and “no additives” as the most important factors from the “internal” criteria group. Notably, “marbling”, “nutritional information”, and “leanness” are five, three, and two-and-a-half times less significant characteristics for Australians than “taste” and “animal welfare” [12].

Next, we analyzed how the choice of pork products among Russian consumers (central region) has changed over the past two years. It was found that Russians are now opting for convenience products such as chilled pork, sausages, and minced meat, while the demand for frozen meat and pork fat has dropped significantly (Figure 1B). The rejection of frozen meat may be due to the wide choice of chilled pork on the shelves. Russian consumers habitually buy pork, they choose fresh and minced meat and sausages. It is noted that Russian consumers prefer to buy chilled pork, minced meat, and bacon, from separate farm shops [19].

Of the key product choice factors for Russian consumers in the minced meat category, according to the Netnography survey, fat content is in third place. The level of customer satisfaction (NPS) with minced pork was 57%, which is significantly higher than for turkey (27%) and chicken (16%) minced meat (Figure 1C). The proportion of the fat content of minced pork in the consumer responses was 12.5%, which was significantly higher than the choice factors for turkey (5%) and chicken (3.4%). The study showed that minced pork has the most positive consumer attitudes out of all categories of minced meat and the lowest level of negative assessments of the type and taste of the product. At the same time, the main consumer complaints revolved around smell and fat content.

However, much of the information about pork consumption today is negative. In Russia, while the consumption of pork in general is increasing (up to 34 kg per capita in 2021), there is a growing tendency towards a reduction in the consumption of fatty meat and backfat in connection with the promotion of a healthy lifestyle.

Our findings are also in line with de Araújo et al. (2021) conclusion: “Consumers positively perceive sensory attributes and recognize meat’s nutritional value, still concerned with fat” [35].

The views of Russian consumers in respect of fatty pork meat are contradictory. On the one hand, they cite its high fat content as a negative factor. On the other hand, the presence of fat and intramuscular fat is welcomed by Russian consumers because it is associated with taste, tenderness, flavor, and juiciness.

Domestic pork, including those of Russian meat and fat breeds, may be preferable for Russian consumers. This tendency has the potential to further preserve local breeds of pigs, as is done in other countries [36]. In the future, it will be interesting to analyze consumer attitudes and intentions in relation to local fat breeds of pigs used for the production of pork and co-products.

Also, unfortunately, it was unable to find information on the impact on the preference for “fatty” and “lean” pork by Russian consumers in relation to the climatic zone where they reside, but there is evidence that people living in “colder regions” prefer fatty meat (for example, there is evidence that Inuit and Eskimo people who eat mostly meat, consuming high levels of saturated fatty acids and cholesterol, lead a healthy lifestyle [37]. In the future, we will endeavor to conduct a study of consumer preferences for pork and backfat among the residents of cold regions in Russia.

The nutrient content of meat, backfat thickness, and fatty acid composition were analyzed from two meat and two meat and fat pig breeds. The chemical compositions of the longissimus muscles of pigs are presented in Table 1. In terms of moisture content, there is a clear division into two groups. The moisture content of pork from meat and fat breeds is 10% lower than that of meat breeds (*p* < 0.05).

The data in Table 1 showed close protein/fat ratios for fatty pig breeds: 1.02 for L and 1.14 for M. The protein content in the longissimus muscle of A is about 1.5 times (*p* < 0.05) higher than in L and M. Although A is bred as a meat breed and meets the respective requirements, D has a significantly higher protein content and a lower fat content than all the breeds studied.

Our data shows a higher IMF content (4.4%) and backfat deposition in Livny pigs than in Duroc (2.7%) and Altai (2.05%) pigs, which is different from commercial cross-bred pigs.

At the same time, measurements of the thickness of the backfat showed, as expected, that it was greater in meat and fat breeds: in Livny—85 ± 14 mm, in Mangalitsa—57 ± 8 mm, in Altai and Duroc—43 ± 7 mm and 35 ± 7 mm (Figure 2A), respectively. The melting temperature of the backfat of A was 40.2 °C, M—38.0 ± 3.8 °C, D—31.6 ± 0.3 °C, and L—35.2 ± 2.6 °C.

The results of the morphological study of backfat showed differences in the cellular structure of the backfat of pigs of the breeds under study. Adipose tissue cells were characterized by dense packing and a polygonal shape. The shape of the cells was more spherical in samples D and A than in L and M. The central part of the cell was occupied by a large fat droplet, displacing the nucleus to the periphery. Thin sparse collagen fibers were located between adipocytes L and M, while intercellular layers of connective tissue in D and A contained more frequent collagen fibers.

The area of adipocytes differed in different layers of the backfat in different breeds (Figure 2B,C).

Pigs of breeds M and A had the smallest average adipocyte sizes in the outer layer, while L pigs had the largest—by 39% and 24% relative to M and A, (*p* < 0.001). Meanwhile, deep layers of subcutaneous fat in samples from all breeds studied revealed adipocytes of 10–30% (*p* < 0.001) greater size than adipocytes in the outer layer. The largest adipocyte sizes in the inner layer of ridge fat were also observed in breeds L and D. These findings are interesting, as the Duroc is considered one of the leanest breeds of pig, and it is expected that backfat tissue will have the smallest adipocyte sizes [38].

Interestingly, M pigs had adipocytes of the same size in the inner layer of the backfat as D pigs. However, meat and fat breeds generally have larger adipocyte diameters than leaner breeds with the same body weight [13]. A “biphasic” distribution of adipocytes into small and large adipocytes was observed in all samples, which is consistent with the results of other studies.

Our study showed that the area of small adipocytes in the outer and inner layers of the backfat was different and varied for L—outer layer [3322.06–4374.59] mkm^2^, inner layer [4110.97–5048.33] mkm^2^; D—outer layer [2741.80–3310.32] mkm^2^, inner layer [2652.09–3667.92] mkm^2^; A—outer layer [2802.54–3326.04] mkm^2^, inner layer [3233.10–3603.68] mkm^2^; M—outer layer [2711.96–2909.65] mkm^2^, inner layer [3291.89–3839.90] mkm^2^. The proportion of small adipocytes was 16.0% and 15.2% for D and A pigs, and 17.3% for L pigs. The highest number of small adipocytes relative to the total number of adipocytes was found in the adipose tissue of M—20.0%. The accumulation of small adipocytes, which subsequently hypertrophied due to triglyceride accumulation, points to an active process of adipocyte hyperplasia, especially in M pigs. This data is consistent with the study [39], which found that in 5-month-old Meishan and Landrace pigs, 15–19% of the total population had small adipocytes in the backfat.

The fatty acid composition results were analyzed to ascertain the SFA, UFA, PUFA and MUFA ratios (Table 2). C8:0 was detected in small amounts only in backfat M, while the highest levels of C12:0 and C14:0 were observed in both backfat L and M. The lowest level of C15:0 was revealed in backfat D, on the contrary the contents of C18:0 and C20:0 exceeded their levels in other breeds by more than 1.4-fold (*p* < 0.05). Backfat L was characterized by the highest concentration of C17:0 and C22:0, while the lowest content of C20:0 was detected in A. The highest content of saturated fatty acids was observed in backfat D—18.1% (*p* < 0.05) higher than in A, while the content of unsaturated fatty acids, in contrast, was 10.7% (*p* < 0.05) lower than in A. C14:1 was detected in small amounts in backfat L and M, as well as the highest level of C16:1, while the lowest content of C20:1 n–9 was detected in A. The levels of C17:1 and C18:1 in backfat L exceeded its levels in other breeds by more than 1.9-fold (*p* < 0.05) and 12.3% (*p* < 0.05), respectively. The content of monounsaturated fatty acids was the highest in backfat L—43.2% (*p* < 0.05) higher than in A, while the content in backfat D was also high, 24.9% (*p* < 0.05) higher than in A.

The level of C18:2 n−6 in backfat A exceeded its levels in other breeds by more than 1.6-fold (*p* < 0.05). C20:5 n−3 and C22:6 n−3 were detected in small amounts only in backfat M, while C18:2 n−6 (linolelaidinic) was were observed in both backfat L and M. The levels of C18:3 n−3 and C20:3 n−3 in backfat M exceeded their levels in other breeds by more than 1.4-fold (*p* < 0.05) and 1.9-fold (*p* < 0.05), respectively. Backfat A was characterized by the highest concentration of C20:4 n−6, C20:3 n−6, and C20:2 n−6, which were higher than in other breeds by more than 1.7-fold (*p* < 0.05), 1.7 fold (*p* < 0.05) and 1.4 fold (*p* < 0.05), respectively. The content of polyunsaturated fatty acids in backfat D and L was 2.0 times (*p* < 0.05) lower than in A, which resulted in the lowest ΣPUFA/ΣSFA ratio—2.5 times (*p* < 0.05), lower than in A. The highest content of n-3 fatty acids was observed in M at 2.6 (*p* < 0.05) and 2.5 times (*p* < 0.05) higher than in D and L, respectively.

The content of n-6 fatty acids in the backfat of D and L was more than 2.0 times (*p* < 0.05) lower than that of A. The highest Σn-3 PUFA/Σn-6 PUFA ratio was observed in the backfat of M at 1.5 to 2.5 times (*p* < 0.05) higher than in the other breeds. The highest ratio of C 18:1/C 14 was observed in the backfat of D, exceeding the lowest ratio observed in M by 48.4% (*p* < 0.05), while the ratio of C 18:2/C 14 in the backfat of D and L was more than 2.0 times (*p* < 0.05) lower than in A and reached 3.4 times (*p* < 0.05) lower.

The highest content of short-chain fatty acids was observed in the backfat of M and was 1.6 times (*p* < 0.05) higher than the content in L. The highest content of medium-chain fatty acids was observed in the backfat of L and M and was 16.2% (*p* < 0.05) and 19.2% (*p* < 0.05) higher than the content in A, respectively. The content of long-chain fatty acids in the backfat of L and M pigs was lower than that of A by 5.4% (*p* < 0.01) and 6.5% (*p* < 0.05), respectively. The highest ΣC4-C16/ΣC17-C24 ratio was also observed in the backfat of L and M.

The lowest atherogenicity index was observed for the backfat of A and was lower than in other breeds by more than 1.2-fold (*p* < 0.05), while the highest thrombogenicity index was determined for backfat of D exceeded its level in other breeds by more than 1.2 fold (*p* < 0.05). The lowest atherogenicity to thrombogenicity index ratio was observed in the backfat of D, which was lower than in other breeds by more than 12.0% (*p* < 0.05).

Thus, we found that raw meat from the Duroc pig, which is bred in Russia and considered one of the leanest breeds of pigs, had the most “lean” intramuscular fat and a thin layer of backfat. At the same time, differences in the cellularity and morphology of backfat adipocytes and fatty acid analysis were found. In the backfat adipocytes of considerable size in the internal layer of backfat were observed, while the number of small adipocytes, which amounted to about 16% of the total number of cells, was approximately the same in the external and internal layers of the samples. This data, together with the high saturated and monounsaturated fatty acid content and high oleic (C18:1) to myristic (C14:0) fatty acid ratio, indicates a higher rate of lipid accumulation and an active process of white adipose fat hypertrophy with fewer but larger cells [40,41]. The backfat of D pigs showed the highest atherogenicity and thrombogenicity indices (0.5 and 1.4).

At the same time, pork from the local Livny pigs was characterized by a high intramuscular fat content, the backfat was the thickest, adipocytes were characterized by a larger area in the inner and outer layers of the backfat, indicating high adipose tissue development and completion of hyperplasia and hypertrophy. The highest mono-unsaturated and medium-chain fatty acid content and the lowest short-chain fatty acid content were observed in the backfat. The ratio of omega 3 to omega 6 was (0.07), with a minimum linoleic (C18:1) to myristic (C14:0) FA ratio (0.58), and the atherogenicity and thrombogenicity indices were 0.5 and 1.2. This data is consistent with the work of [15], which reported that high intramuscular and backfat values correlated with an increase in the percentage of monounsaturated fatty acids.

According to our data, the highest UFA content, in particular omega 3 and omega 6 PUFA, was found in the backfat of Altai meat breed pigs with a minimum SFA content. At the same time, a shift in the long-chain FA profile towards an increase in C18:3 ώ3 and a decrease in the proportion of C18:2 ώ6 FA, which, according to [42] facilitates the endogenous synthesis of eicosapentaenoic and docosahexaenoic ώ3 FA.

The most interesting results were obtained for the meat and backfat of Mangalitsa pigs raised in Eastern Siberia under free-range conditions. With a significant amount of intramuscular fat, the thickness of the backfat of this pig breed was comparable to that of the Altai meat pig. With a significant number of small adipocytes (relative to the total number of adipocytes—20.0%), a significant variation in their area for the outer and inner layer was observed, pointing to the possible dynamic cellularity of white adipose fat. In terms of the fatty acid composition of backfat, high values of omega-3 fatty acids and short-chain fatty acids were found.

The ∑PUFA/∑SFA saturation index has significant differences in the fat of the selected pig breeds. The highest index in the backfat of A is 0.77 ± 0.04, while in the D, M, and L, it is 0.30 ± 0.08, 0.46 ± 0.02, and 0.28 ± 0.09, respectively. Meanwhile, the best n-3 PUFA/n-6 PUFA ratio was observed in the M (0.103 ± 0.01), which can be explained by the free-range rearing and diet of these pigs [43].

The Raman spectra of pig fat samples were mainly represented by bands due to oscillations of hydrocarbon chains (saturated and unsaturated structures) (Figure 3). The peaks at 970 cm^−1^, 1266/1272 cm^−1^, and 1655 cm^−1^ reflect the degree of unsaturation, while the signals at 1297/1301 cm^−1^, 1430/1460 cm^−1^ and 1735/1746 cm^−1^ correspond to saturated bonds or ester groups [28,44]. The mean intensities were 0.529505 for A; 0.452598 for M; 0.37439 for L; and 0.36496 for D.

Values of peak areas and their intensities are presented in Appendix A. The peak area in the region of 960–980 cm^−1^ for backfat of A was higher than in other pig breeds (by 2.2-, 2.0-, and 1.5-fold than in D, L, and M respectively, *p* < 0.05). The intensity of this peak was higher in both A and D backfat than in L and M ones by more than 1.7-fold (*p* < 0.05). In the backfat of A, the peak in the region 1260–1280 cm^−1^ was characterized by the highest area and exceeded the L value by 3.5 times (*p* < 0.05). The peak intensity was the highest in backfat D—1.7 fold (*p* < 0.05) higher than in L. The peak area in the region of 1290–1310 cm^−1^ in the backfat of D was more than 1.3 times (*p* < 0.05) lower than that of L and M, as well as its intensity was lower than L by 11.1% (*p* < 0.05). There was no statistical difference in peak areas and their intensities in the region of 1420–1470 cm^−1^. The peak areas were not statistically different in the region of 1720–1760 cm^−1^, but the intensity of this peak was higher in both A backfat than in D by 1.2-fold (*p* < 0.05).

Moderate interaction between peak areas at 1290–1310 sm^−1^, which are markers for the FA saturation degree, and the saturated FA content in D fat (r = 0.58, df = 13, *p* < 0.05, Pearsen) was revealed. Similar results were obtained for A fat while comparing peak areas at 1720–1760 sm^−1^ and the sum of unsaturated fatty acids (r = 0.53, df = 13, *p* < 0.05, Pearsen).

These findings are consistent with those of the FA composition study, and the work of [45,46], in which correlations (R CV 2 = 0.8–0.9) were obtained between the Raman spectra and the overall FA composition parameters (SFA, MUFA, and PUFA), determined by gas chromatography.

From the data obtained it was revealed that the backfat samples from Altai meat pigs contained more unsaturated fatty acids than from other breeds. The highest amount of saturated fatty acids was found in backfat samples from the Duroc breed.

In conclusion, it is worth pointing out the importance of preserving local breeds to facilitate diet diversification, which is important for overall health. However, much of the information about pork consumption today is negative. In Russia, while the consumption of pork in general is increasing (up to 34 kg per capita in 2021), there is a growing tendency towards a reduction in the consumption of fatty meat and backfat in connection with the promotion of a healthy lifestyle.

## 4. Conclusions

Our research has shown that the Russian consumer prefers chilled pork either as a whole piece or as minced meat. At the same time, the Russian consumer, similar to consumers elsewhere, is concerned about fatty pork as a potentially unhealthy product. At the same time, unlike the Western consumer, the Russian one is ready to buy pork with a fat content of over 10%, associating it with better taste characteristics such as juiciness, tenderness, and aroma.

It was shown that pork of local breeds is characterized by better IMF content, the share of which is sufficient to attract a Russian consumer.

The Livny pigs differed significantly from the other studied breeds in terms of a range of indicators. According to the structure of the backfat and the composition of fatty acids, the meat and fat of Livny pigs can be considered sources of compounds, which in benefit ratio could provide health-promoting capability.

This means that pork can be part of a healthy dietary pattern if consumed rationally, and there is no reason to demonize this valuable product from a medical point of view. It is important to educate the public as well as the food industry and medical community. The rational consumption of pork and backfat, its combination with other foodstuffs, and the creation of new products enriched with biologically active substances is possible with a thorough understanding of its composition, including breed-specific features.

## Figures and Tables

**Figure 1 foods-12-00690-f001:**
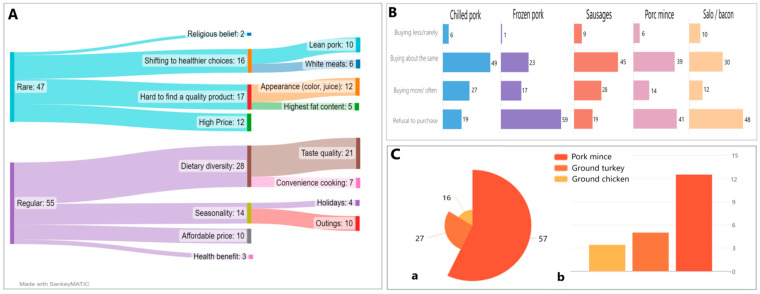
Russian consumer attitude towards pork attributes: (**A**) Russian consumer motives for choosing pork (%); (**B**) Consumer attitudes towards pork products bought over the past 1–2 years, % of respondents (survey ^1^); (**C**) The Russian consumers preferences for minced meat: (**a**) Net Promoter Score; (**b**) Percentage of mentions of fat content. Note: ^1^ Ponamoreva E. Trends Laboratory (LLC Business Integrator Territory of Success) & Agro-Belogorye, conducted in March 2022 [19]. Source: Authoring.

**Figure 2 foods-12-00690-f002:**
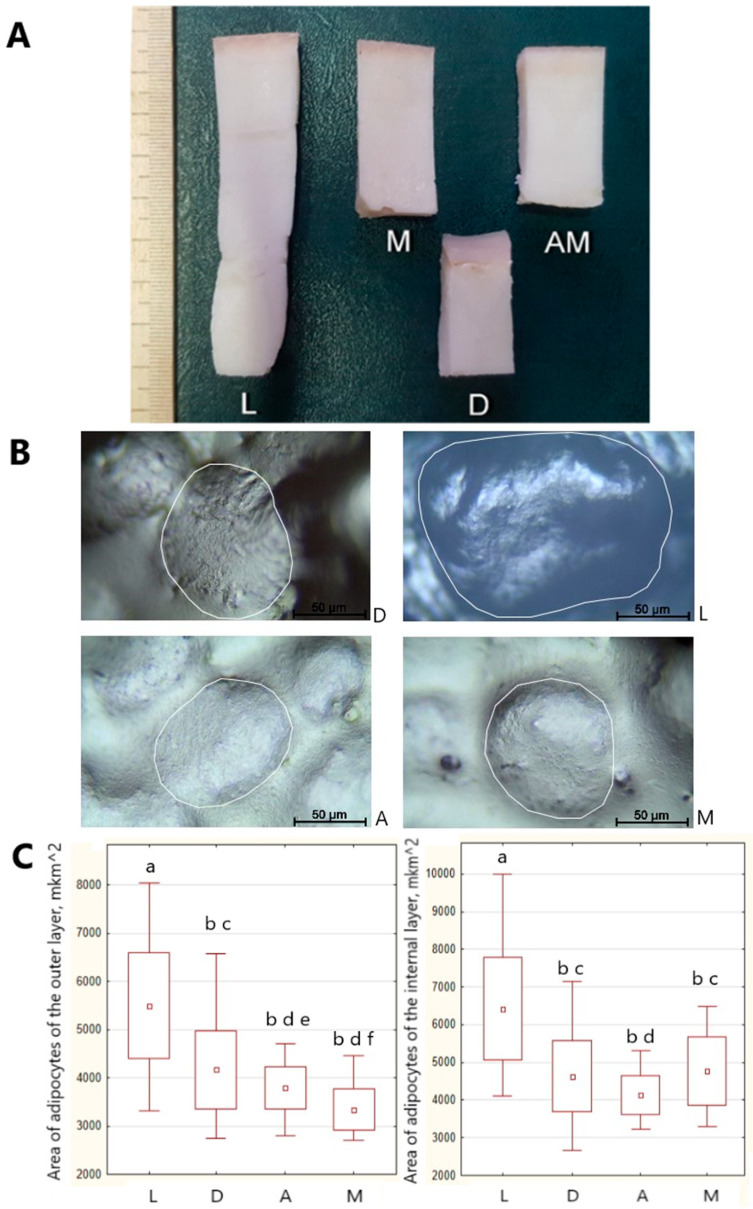
Backfat depending on pig breed: (**A**) The thickness of the backfat; (**B**) Histomorphological traits of backfat; (**C**) The area of adipocytes in different layers of the backfat: Data presented as box plots illustrate hinges extending from the mean ± SD, the mean point within the box and whiskers extending to the minimum and maximum values of the dataset. Note: a-b, c-d, e-f—Different letters indicate statistically significant differences (*p* ≤ 0.05).

**Figure 3 foods-12-00690-f003:**
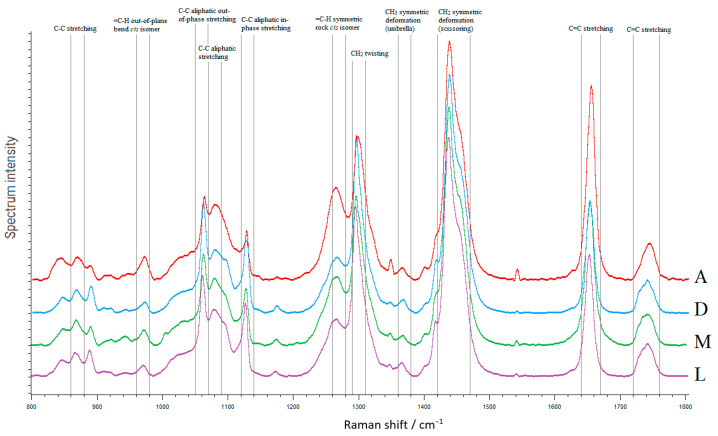
Raman spectra of backfat samples from four pig breeds, detection range 800 and 1800 cm^−1^.

**Table 1 foods-12-00690-t001:** Proximate composition of longissimus muscle, % to dry matter (Mean ± SD).

Items (%)	Pig Breed			
	Livny	Duroc	Altai	Mangalitsa
Moisture	62.9 ±5.0 ^a^	71.88 ± 1.01 ^b^	70.22 ± 0.76 ^b^	63.45 ± 0.65 ^a^
Protein	18.23 ± 2.71 ^a^	22.72 ± 0.14 ^b^	22.10 ± 0.52 ^b^	18.91 ± 1.13 ^a^
Fat	17.92 ± 1.40 ^a^	4.32 ± 0.13 ^b^	6.40 ± 0.38 ^b^	16.64 ± 2.12 ^a^
Ash	0.95 ± 0.13	0.96 ± 0.05	1.12 ± 0.05	0.95 ± 0.09

Note: a, b—Different letters indicate statistically significant differences (*p* ≤ 0.05).

**Table 2 foods-12-00690-t002:** Results of fatty acid composition (mass fraction, % of total fatty acids) of backfat samples of different species (Mean ± SD).

Parameters	Duroc	Livny	Altai	Mangalitsa
C8:0	0.00 ± 0.00 ^a^	0.00 ± 0.00 ^a^	0.00 ± 0.00 ^a^	0.01 ± 0.01 ^b^
C10:0	0.06 ± 0.03	0.06 ± 0.01 ^a^	0.07 ± 0.01	0.09 ± 0.00 ^b^
C12:0	0.07 ± 0.01 ^a^	0.09 ± 0.02 ^b^	0.08 ± 0.01 ^a^	0.10 ± 0.01 ^b^
C14:0	1.21 ± 0.18 ^a^	1.67 ± 0.23 ^b^	1.28 ± 0.06 ^a^	1.63 ± 0.13 ^b^
C15:0	0.04 ± 0.03 ^a^	0.09 ± 0.02 ^b^	0.07 ± 0.01 ^b^	0.06 ± 0.02
C16:0	24.16 ± 1.32 ^a^	24.11 ± 2.01 ^a^	21.76 ± 0.35 ^b^	25.00 ± 1.01^a^
C17:0	0.70 ± 0.09 ^a^	1.07 ± 0.20 ^b^	0.86 ± 0.11a	0.72 ± 0.02 ^a^
C18:0	17.31 ± 2.59 ^a^	12.39 ± 1.90 ^b^	12.81 ± 1.47 ^b^	12.58 ± 1.07 ^b^
C20:0	0.29 ± 0.05 ^a^	0.18 ± 0.02 ^b^	0.21 ± 0.04 ^b^	0.14 ± 0.12 ^b^
C21:0	0.10 ± 0.03	0.11 ± 0.12	0.04 ± 0.01	0.09 ± 0.05
C22:0	0.01 ± 0.02 ^a^	0.05 ± 0.01 ^b^	0.02 ± 0.02 ^a^	0.01 ± 0.01 ^a^
ΣSFA	43.93 ± 2.93 ^a^	39.82 ± 3.91	37.18 ± 1.68 ^b^	40.42 ± 1.79
C14:1	0.00 ± 0.00 ^a,c^	0.03 ± 0.01 ^b^	0.00 ± 0.00 ^a,c^	0.02 ± 0.01 ^a,d^
C16:1	2.03 ± 0.63 ^a^	3.15 ± 0.47 ^b^	1.90 ± 0.31 ^a^	3.11 ± 0.42 ^b^
C17:1	0.26 ± 0.06 ^a^	0.49 ± 0.11 ^b^	0.23 ± 0.15 ^a^	0.26 ± 0.02 ^a^
C18:1	39.50 ± 1.61 ^a^	44.37 ± 1.87 ^b,c^	31.55 ± 2.23 ^b,d,e^	36.15 ± 1.82 ^b,d,f^
C20:1 n–9	1.13 ± 0.25 ^a^	1.15 ± 0.23 ^a^	0.68 ± 0.03 ^b^	1.16 ± 0.02 ^a^
C22:1 n–9	0.01 ± 0.02	0.02 ± 0.02	0.02 ± 0.03	0.02 ± 0.03
ΣMUFA	42.91 ± 1.68 ^a,c^	49.20 ± 2.15 ^b^	34.37 ± 2.57 ^a,d^	41.04 ± 1.91 ^a,c^
C18:2 n−6	11.44 ± 2.39 ^a,c^	9.39 ± 2.08 ^a,c^	25.65 ± 1.35 ^b^	15.64 ± 0.43 ^a,d^
C18:2 n−6	0.00 ± 0.00 ^a^	0.03 ± 0.02 ^b^	0.00 ± 0.00 ^a^	0.04 ± 0.04 ^b^
C18:3 n−3	0.60 ± 0.32 ^a,c^	0.60 ± 0.15 ^a,c^	1.01 ± 0.07 ^b^	1.42 ± 0.21 ^a,d^
C20:4 n−6	0.21 ± 0.06 ^a^	0.23 ± 0.09 ^a^	0.39 ± 0.06 ^b^	0.22 ± 0.05 ^a^
C20:5 n−3	0.00 ± 0.00 ^a^	0.00 ± 0.00 ^a^	0.00 ± 0.00 ^a^	0.01 ± 0.02 ^b^
C20:3 n−6	0.09 ± 0.02 ^a^	0.10 ± 0.03 ^a^	0.17 ± 0.02 ^b^	0.09 ± 0.05 ^a^
C20:2 n−6	0.75 ± 0.15 ^a,c^	0.53 ± 0.21 ^b^	1.10 ± 0.07 ^a,d^	0.80 ± 0.02 ^a,c^
C20:3 n−3	0.07 ± 0.05 ^a^	0.10 ± 0.05 ^a^	0.13 ± 0.01 ^a^	0.24 ± 0.05 ^b^
C22:6 n−3	0.00 ± 0.00 ^a^	0.00 ± 0.00 ^a^	0.00 ± 0.00 ^a^	0.07 ± 0.06 ^b^
ΣPUFA	13.16 ± 2.82 ^a,c^	10.97 ± 2.55 ^a,c^	28.45 ± 1.47 ^b^	18.54 ± 0.49 ^a,d^
ΣUFA	56.07 ± 2.93 ^a^	60.18 ± 3.91	62.82 ± 1.68 ^b^	59.58 ± 1.78
ΣPUFA/ΣSFA	0.30 ± 0.08 ^a,c^	0.28 ± 0.096 ^a,c^	0.77 ± 0.04 ^b^	0.46 ± 0.02 ^a,d^
Σn−3 PUFA	0.67 ± 0.36 ^a,c^	0.70 ± 0.19 ^a,c^	1.13 ± 0.08 ^b^	1.74 ± 0.20 ^a,d^
Σn−6 PUFA	12.49 ± 2.55 ^a,c^	10.27 ± 2.37 ^a,c^	27.31 ± 1.41 ^b^	16.81 ± 0.30 ^a,d^
Σn−3 PUFA/Σn−6 PUFA	0.05 ± 0.02 ^a^	0.07 ± 0.01 ^a,c^	0.04 ± 0.00 ^a,d^	0.10 ± 0.01 ^b^
C18:2/C14	9.53 ± 1.82 ^a,c^	5.89 ± 2.28 ^b^	20.04 ± 1.31 ^a,d^	9.62 ± 0.55 ^a,c^
C18:1/C14	33.37 ± 4.91 ^a^	27.14 ± 4.88 ^b^	24.65 ± 1.97 ^b^	22.48 ± 2.87 ^b^
ΣFAshort (from C4 to C10)	0.062 ± 0.032 ^a^	0.059 ± 0.009 ^a^	0.067 ± 0.011	0.093 ± 0.010 ^b^
ΣFAmedium (from C11 to C16)	27.49 ± 1.81 ^a^	29.14 ± 2.40 ^a^	25.09 ± 0.12 ^b^	29.91 ± 1.30 ^a^
ΣFAlong ( > C17)	72.45 ± 1.83	70.80 ± 2.41 ^a^	74.85 ± 0.13 ^b^	70.00 ± 1.29 ^a^
ΣC4-C16/ΣC17-C24	0.380 ± 0.03	0.414 ± 0.047 ^a^	0.336 ± 0.002 ^b^	0.429 ± 0.026 ^a^
IA	0.520 ± 0.047 ^a^	0.517 ± 0.078 ^a^	0.430 ± 0.015 ^b^	0.532 ± 0.039 ^a^
IT	1.44 ± 0.19 ^a^	1.21 ± 0.21 ^b^	1.05 ± 0.08 ^b^	1.15 ± 0.08 ^b^
IA/IT	0.362 ± 0.031 ^a^	0.430 ± 0.017 ^b,c^	0.411 ± 0.017 ^b,c^	0.464 ± 0.023 ^b,d^

Note: a-b, c-d, e-f—Different letters indicate statistically significant differences (*p* ≤ 0.05).

## Data Availability

Data is contained within the article or Appendix A.

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
