# Peer review of "Pork Fat and Meat: A Balance between Consumer Expectations and Nutrient Composition of Four Pig Breeds"

_foods, 2023, doi:10.3390/foods12040690_

Round 1
Reviewer 1 Report
Comments and Suggestions for Authors
An attempt is made in the article to present a comparative study of consumer evaluation and purchasing behavior of Russian consumers of fatty meat and fat from four pig breeds (A) Altai (meat), (L) Livni (meat and fat), (M) Russian Mangalitsa and (D) Russian Duroc.
As methods of analysis are used: Netnographic studies of consumer purchasing behavior; the proximate composition of representatives of muscle and fat tissue from the four pig’s breeds; Raman spectroscopic and histological analysis of fats.
According the authors the results obtained demonstrated an inconsistency in the views of Russian consumers, who perceive high fat content as a negative factor, but at the same time approve of the presence of intramuscular fat in meat, associating it with more acceptable meat taste, greater tenderness, aroma and juiciness.
The conclusion was made that pork from local Russian breeds can be recommended as a functional food in the context of the requirement to change the strategy to promote the consumption of local pork from the standpoint of dietary diversity and health.
Improvement opportunities
Title, Objectives and Structure of the Article
The title and stated purpose of the study are not an emanation of the results obtained and the conclusions drawn.
From the results presented in the manuscript, it is perhaps more appropriate that the title should be directed to a comparative study of the nutritional value of fat from four suborders of pigs in the Russian Federation.
It was found that the aim of the study stated in line 10 of the Abstract does not exactly match the one written on lines 63-65 at the end of the Introduction. I ask the contributors to be consistent and to formulate the aim of the study in the same way both in the abstract and in the text.
It is more than natural that the title, the methods of analysis used, the results obtained and the conclusions drawn derive directly from the set purpose of the study. Unfortunately, this is not exactly the case in this manuscript.
The main objective was achieved after reviewing literature sources (see section 2.1 Marketing research - lines 68-93 and 169-245 of section 3. Results, incl. and F ig. 1, which does not present its own data, but cites other authors. From the presented material, it is not clear whether the author collective received consent from the authors of the presented figure to publish it in this study of theirs?), rather than using modern methods of consumer preference research, marketing research or sensory analysis.
In this sense, I suggest that the authors consider whether it is not more appropriate to change the type of the manuscript from a research article to a review article with elements of own research.
Therefore, I recommend the authors to discuss changes and the structure of the manuscript, which needs significant improvement.
Abstract
If the authors and the Editorial Board accept the proposed changes in the type and structure of the manuscript, should the abstract be fundamentally revised?
The submitted (4) Conclusions – lines 19-20 of the Abstract, that: "The results show that pork from local breeds can be recommended for functional foods." do not derive from the results presented in this study. There is no evidence that the fatty acid composition of porcine adipose or muscle tissue has been modelled or modified by either in vivo or in vitro experiments. In this sense, the fat or meat does not meet the definitions for formulating functional foods! Perhaps the authors wanted to emphasize the possibility that the adipose tissue from pigs thus studied could be offered as suitable for rational consumer nutrition?
1. Introduction
The introduction should be structured in accordance with the last specified aim of the study and with the structure of the manuscript.
I suggest that the authors to consider moving of some paragraphs from the Results and Discussion sections to the Introduction, because they neither comment on the presented results nor provide a theoretical explanation of the results when discussing them.
Such are the following texts:
Lines 187-189 - We tried to understand whether this increase in consumption is only due to socio-economic reasons, or whether it is also due to some kind of consumer preference for pork.
Lines 203-210 – It is important to highlight another factor influencing the consumption of pork – the growing trend towards the importance of animal welfare and care for the environment. This trend is gaining momentum especially among younger generations [16, 17], as well as high-income consumers who are switching to a diet that reduces or eliminates the consumption of meat and meat products and redistributes the basket towards white/lean meat. However, research shows [18] that the proportion of consumers worldwide that is prepared to stop or significantly reduce their consumption of meat to protect the environment is low.
Lines 232-236 - In a study [22] changes in consumer preferences were correlated with a change in lifestyle promoted by public health bodies by reducing red meat consumption and moderate fatty pork consumption. Consumers habitually buy pork, they choose fresh and minced meat, and the most often buy locally sourced meat from butchers.
Lines 246-249 - Given that meat fat content is directly related to fatty acid composition, which in turn is closely related to fat deposition and hence to intramuscular fat content and backbone fat thickness [4], we then analysed these indices for two meat and two meat and fat pig breeds.
Lines 366-392 - The key factor in shaping consumer attitudes towards a product is information. The nature of information can be positive, describing benefits, negative, listing risks, or balanced. However, negative information is more persistent than positive information and is related to the instinctive desire of consumers to avoid products for which there is negative information [28, 29]. According to Hamlin (2016) [30], the shaping of consumer opinions about a product can be divided into two types: functional and constructive attitudes. While the former remains in the consumer's memory for a long time, the second group of motivations is created in situ to make a single decision in response to specific circumstances.
Pork can no longer be regarded as a commodity. In Russia, separate consumer segments with different tastes and opinions on the quality and content of pork and pork products are being shaped. A large number of consumers belong to the group who prefer pork to other meats "because they like pork" and would like it to have health benefits too.
There is a preconception that backfat from 'lean' meat pigs may be more beneficial for health.
By distinguishing between SFA, MUFA and PUFA groups, it is advisable to ascertain not only the impact on the physiological processes of each group and their absolute content in different products, but also their ratio. The ∑PUFA/∑SFA index is used to show atherogenicity. Its use indirectly assesses the impact of diet on cardiovascular health, and is based on the recognised hypothesis that all unsaturated FA can suppress LDL and contribute to lower blood cholesterol levels, and saturated FA vice versa. This means that the higher the ∑PUFA/∑SFA index, the greater the health benefits of the diet.
Also, to predict the impact on health, nowadays attention is paid to the n-3 PUFA/n-6 PUFA ratio, which is skewed towards n-6 in pork, which is not in line with current dietary recommendations. However, the 0.2 ratio recommended by nutritionists needs to be maintained in the overall dietary pattern, hence when including pork in the diet, this ratio should be adjusted at the expense of foods high in n-3.
Lines 438 - 443 - In conclusion, we would like to point out the importance of preserving local breeds to facilitate diet diversify, which is important for overall health. However, much of the in-formation about pork consumption today is negative. In Russia, while the consumption of pork in general is increasing (up to 34 kg per capita in 2021), there is a growing tendency towards a reduction in the consumption of fatty meat and backfat in connection with the promotion of a healthy lifestyle.
There is some inaccuracy in line 55 - The fat content of pork, like any product, is now viewed as a complex of fatty acids that play both positive and negative roles in the prevention of non-communicable diseases. It is probably about metabolic diseases such as atherosclerosis, myocardial infarction, stroke, obesity, type 2 diabetes, etc.
2. Materials and Methods
I suggest the authors to consider transfer the following paragraphs from section 3. Results to section 2.1 Marketing research, because they practically describe the research method and do not discuss specifically obtained results to:
Lines 170 - 179 - During the analysis of Russian documents that matched the search characteristics of our study, both for pork and for variables related to purchasing behaviour, a total of 107 documents were found in the first stage. Of these, after sifting out duplicates that did not fit the empirical article, matching the exclusion criteria, only 15 studies remained for analysis. At the same time, 3 regions were found in which studies were conducted, which were distributed as follows: Central Federal District (Moscow – 4, St. Petersburg – 3), Siberian Federal District (Altai Krai – 3, Kemerovo Oblast – 2), Southern Federal District (Rostov Oblast – 2). Based on the data presented in the articles, we categorised the responses into "rarely" and "often", shaping the factors that determine consumer assessments.
I suggest the authors to consider supplementing the following information in section 2.2 Animals, diets and sample preparation
- To add clarifications as to whether the experiment was conducted in accordance with Art. 14 of Part V. Breeding and Animal Husbandry Units of the European Convention for the Protection of Vertebrate Animals Used for Experimental and Other Scientific Purposes, Commission Recommendation 2007/526/EC and Council Regulation (EC) No. 1099/2009. Whether the experiment has been approved by the Russian Commission on Scientific Ethics (if any) and whether the requirements of Council Directive 2010/63/EU have been met.
- To give the explanation and to make the appropriate additions conncted with too limited use of animals of one breed 5-6 pcs is described. Is this enough to guarantee the statistical reliability of the results? Exactly how many animals does each group contain?
- To describe whether female or male pigs or both female and male pigs were used in the experiment? If so, what is the ratio of male to female animals?
- To report the mean live weight (mean ± SEM) of each group of slaughtered animals]
-To indicate the age at which the animals were slaughtered!
- To add information on how the pigs were raised - were they housed in a barn equipped with individual boxes with feeders and drinkers or not?
- Where are the pigs raised? During the experiment, what was the temperature of the air because it affects the tenderization and chemical composition of the meat and fat tissue?
- Was feed available ad libitum or not? Was water provided by nipple drinkers ad libitum or otherwise - unclear?
- Were the pigs randomized by origin, age, sex, and weight?
- Is there a control group against which the comparative studies were made?
- How long was the feeding of the animals (if a feed finisher was used - for how long was it applied?);
- To specify the chemical composition (proximate composition) of the diet of each of the four groups of pigs, because it has a significant effect on the proximate composition and fatty acid composition of the tissues and on the nutritional value of the pork and bacon produced. Please add this information in tabular form, as well as macronutrients and micronutrients - minerals, vitamins, etc.
It is desirable in section 2.3 Proximate analysis to briefly describe the principles of the methods used for the determination of moisture, protein, lipids and ash according to which AOAC method [8]?
In section 2.4 Chromatography of fatty acids, the source that proposed the method of analysis is not cited! Is this a proprietary method of the author collective?
On line 126 is used the terminology “Classic indices such as…”. Isn't it more accurate to talk about Nutritional indices for fatty acids assessing?
I recommend that the authors discuss a change in the title of section 2.5 Histological data. Isn't it more correct to report on 2.5 Histological analysis?
I suggest the authors consider removing the text form lines 353 – 356 – “To determine the relative content of unsaturated fatty acids in each sample, we calculated the intensity ratios of the Raman light scattering signals corresponding to unsaturated bonds to those of saturated or ester bonds. We used nine intensity ratios: I970/I1297, I970/I1430, I970/I1735, I1266/I1297, I1266/I1430, I1266/I1735, I1655/I1297, I1655/I1430, I1655/I1735.” in section 2.6 Raman spectroscopy.
Section 2.7 Statistical analysis needs significant revision with additions and clarifications.
The authors report that the results were obtained from at least three replicate experiments. Please specify exactly how much in each of the experiments, because with three repetitions of biological experiments with pigs, no reliability of the statistical processing of the data is guaranteed!
The text presents data on the melting temperature of back fat, but a method for its determination is not cited and described in section 2. Materials and Methods!
Specify exactly where the mean (M) and standard error (SE) or where the median (Me) and standard deviation (SD) are selected for calculation?
I recommend to standardize the approach and to present all the presented results in terms of the mean (M) and the standard error of the method (SEM). Please cite the source of statistical processing as well as the research method - is it ANOVA or other? Please state appropriately which of the means are significantly different and at what probability is the null hypothesis accepted or what is the confidence level?
Cite in References the non-parametric Kruskal-Wallis test. Please explain how come the authors choose a probability of 0.1 as the significance level? Is it not a consequence of the statistical data analysis method used?
3. Results and 4. Discussion
I recommend that the authors merge Sections 3. Results and 4. Discussion. And to the present manuscript there are attempts to discuss in Section 3. Results: lines 211 - 224 – “The predominant factors influencing the meat purchasing decision process in Norway are taste, ease of preparation and price [18]; in Japan the factors are marbling, an attractive appearance, the absence of negative product information, and local raw materials [19]; in Spain, the consumer is primarily concerned with safety, animal welfare and the environment combined with a hedonistic approach to food. Interestingly, 73% of Spanish consumers surveyed preferred animal friendly production products, but about 70% of consumers are not willing at all or are willing to pay only 5% more for such products. At the same time, 88% were in flavour of products made from local raw materials [20]. Australian consumers ranked taste, juiciness and odour lessness as the three most important factors according to 15 intrinsic and 31 extrinsic criteria in relation to pork meat, and "animal welfare", "country of origin" and "no additives" as the most important factors from the "internal" criteria group. Notably, "marbling", "nutritional information" and "leanness" are five, three and two-and-a-half times less significant characteristics for Australians than "taste" and "animal welfare" [21].”
In the paragraph from lines 180 - 186, the full titles of the literary sources are cited and the corresponding numbers from the list of references are given in middle brackets. This need not be duplicated. I recommend the authors to remove unnecessary texts, for example: "Nationwide statistics show that in 2018, 70.6% of Russians over 14 years of age eat pork daily or several times a week (Results of a 2018 sample dietary survey of the population [13]), with an average of over 20.7 kg per Russian (Macroeconomic review: On food security and consumption of staple foods in Russia. Institute for Comprehensive Strategic Studies [14]). According to preliminary data, in 2021 Russians consume on average 34 kg per year, which is 44% of total consumption (Consumption of pork: Rosselkhozbank Industry Expertise Center [15])".
Figure 1 does not represent results of its own research and should not fall under the Results section. A literary source is cited that is not described in the References. I have a question for the authors, have they obtained the authors' consent to reprint their graphic image and publish their copyrighted results?
Lines 211 - 224 – This text is appropriate to be used in section 4. Discussion – “The predominant factors influencing the meat purchasing decision process Norway are taste, ease of preparation and price [18]; in Japan the factors are marbling, an attractive appearance, the absence of negative product information, and local raw materials [19]; in Spain, the consumer is primarily concerned with safety, animal welfare and the environment combined with a hedonistic approach to food. Interestingly, 73% of Spanish consumers surveyed preferred animal friendly production products, but about 70% of consumers are not willing at all or are willing to pay only 5% more for such products. At the same time, 88% were in flavour of products made from local raw materials [20]. Australian consumers ranked taste, juiciness and odour lessness as the three most important factors according to 15 intrinsic and 31 extrinsic criteria in relation to pork meat, and "animal welfare", "country of origin" and "no additives" as the most important factors from the "internal" criteria group. Notably, "marbling", "nutritional information" and "leanness" are five, three and two-and-a-half times less significant characteristics for Australians than "taste" and "animal welfare" [21].”
Please explain why you presented the data in Table 1 for the muscle Longissimus dorsi? Is it not more correct in this table to present the proximal composition of fats subject to further presentation? In my humble opinion, there is an opportunity here to significantly improve the presented results!
Data from Table 2 is incomplete. Please indicate not only the total percentage of the individual groups of fatty acids, but also indicate the specifically established average value of each fatty acid residue and below indicates the sum as a total. For example, for each sample of lard, what is the content of C 10:0, C 12:0, C 14:0, C 15:0, C 16:0, C 17:0, C 18:0, etc. and underline the sum of saturated fatty acids (SFA), as %. In a similar way to proceed for the group of monounsaturated fatty acids (MUFA) and for polyunsaturated fatty acids (PUFA)!
In the caption of Figure 3. Raman spectra of back fat samples from four pig breeds, a detection range of 700 and 1800 cm-1 is indicated, but in the plot, it starts at 800?
On line 362 a source is cited - Berhe D.T. [27], only [27] can remain. The similar situation is on line 422 - Griffin B.A., 2008 [33] and 444 - Araújo's (2021) – it is appropriate to be de Araújo et al. [35]
The section 5. Conclusions are not formulated in accordance with the set objective of the study.
In practice, these are conclusions to be drawn after each section of discussion of the presented results - by indicators that have been investigated?
The final conclusion should provide information on whether and to what extent the aim of the study was achieved and how this affects the object of the consumer demand (consumer expectations and habits) for pork fat.
Conclusion 6 (lines 472-474) and the last paragraph of the conclusion (lines 479-484) do not follow directly from the results obtained and leave the impression of unfounded speculation.

Author Response
Dear Reviewer 1,
Thank you for valuable comments and recommendations.
Please, find enclosed the answers.
Best regards,
Authors.

Reviewer 2 Report
This study aims to examine trends in consumer attitudes towards the fat content of pork and to assess the prospects for the consumption of fatty pork and backfat from four Russian pig breeds (L, D, A and M). I believe the research is of great value within the scope of the journal. To move forward with the publication, here are some of the concerns that I recommend the authors clearly identify, accordingly.
Lines 32-39
The data presented in this section needs to be supplemented with data sources (literature, websites, or databases).
Lines 163-168
The authors define p<0.1 as a significant difference in the section of “2.7 Statistical analysis”, but mostly discuss the difference at p<0.01 in the results section. It is therefore suggested to revise the definition of p<0.01 as a significant difference.
The proposed statistical method is modified to compare the differences between multiple groups using Duncan's multiple comparisons. Current statistical methods make it confusing and unclear to label the significance of differences in tables and graphs.
In Table 1, it is not clear whether the significance marker for the difference is forgotten, or if there is no significant difference. If there were no significant differences observed, it would not make sense to describe the results as "lower water content (lines 251-252)," "higher protein content (lines 257-258)," "lower fat content (lines 258-259)," and "higher intramuscular fat content (lines 260-264).
Some of the data in Table 2 have significance markers and some do not. This is hard to follow, so I suggest using Duncan's multiple comparisons to compare the differences between multiple groups.
Line 255, the authors wrote as “The data in Table 1 showed close protein/fat ratios for fatty pig breeds: 1.02 for L and 1.14 for M.” However, the relevant data are not shown in Table 1. Please replenish if necessary.
It is recommended that the survey information material covered in Figure 1 (survey methodology/questionnaire/data collected, etc.) be described in detail in the supplementary material. I believe this will go a long way to further understanding the findings of the survey.
The background in Fig. 2B is not uniform, making it difficult to see and distinguish the difference. Moreover, Fig. 2C needs to be corrected for the high-definition image version.
In Figure 3, it is suggested to add a bar chart to more visually compare the unsaturated fatty acid profiles of different breeds of pigs.
The logical lines of the paragraphs are not clear enough and need to be reorganised so that sections of short paragraphs separated in logical unity can be grouped together, especially in the introduction and results sections.
The conclusion part (lines 454-478) is very wordy and needs to be condensed and simplified.
Overall, I think the authors need to take some time and effort to carefully revise the manuscript.
Author Response
Dear Reviewer 2,
Thank you for valuable comments and recommendations.
Please, find enclosed the answers.
Best regards,
Authors.

Reviewer 3 Report
(1)Line 15-17. “Results: the inconsistency of Russian consumer views was noted: high fat content is a negative factor, but intramuscular fat is welcomed, since it is associated with a better meat taste, tenderness, aroma and juiciness.” This is your key result or finding, right? But I don't think such a statement is a result statement, the result should be a fact description, don't add since,,, etc
(2)Line 19-20. “The results show that pork from local breeds can be recommended for functional foods” functional foods? What is the definition of functional foods? This sentence is inappropriate as a conclusion
(3)Line 30-31. “More than half of this increase will come from increased per capita consumption of poultry meat” Your topic is pork, so it is not appropriate to talk about poultry, remove
(4)Line 48-53. “There is a discrepancy in consumer opinion between what product they want to eat, what they think they eat and what they actually eat. A clear illustration of this observation is consumer attitude to pork fat. On the one hand, the fat content of pork may be a factor that can potentially lead to a reduction in the appeal of a product. On the other hand, consumers in many countries do not consider the fat content of pork as a significant factor when choosing a product in the supermarket.” This description is very interesting, do you have a reference, especially some countries where consumers do not consider the fat content of pork
(5)Line 60-66. These two paragraphs should be combined
(6)Line 69-73. “Russian scientific digital library sources (elibrary.ru), portals (https://meat-expert.ru/), statistical materials (https://rosstat.gov.ru/) from 2017 to 2022 were used to search for articles. Default terms and the logical connectors "AND" and "OR": "pork" AND ((purchasing behaviour OR purchase intent) OR (customer satisfaction) OR (consumer ratings) OR (consumer perception) OR (customer satisfaction)) were used to select studies.” I would like to know the time of the search, and also why choose this way to search?
(7)Line 232-236. “In a study [22] changes in consumer preferences were correlated with a change in lifestyle promoted by public health bodies by reducing red meat consumption and moderate fatty pork consumption. Consumers habitually buy pork, they choose fresh and minced meat, and the most often buy locally sourced meat from butchers.” What is the purpose of this description?
(8)Table1. Why is there such a big difference between protein and fat content, and when measuring it, is the same part selected for measurement or is it different meat? In addition, how many repetitions need to be marked.
(9)Figure.2 C. This picture resolution is not clear
(10)Figure 3. Raman spectroscopy should be more analyzed and discussed, which is what makes your article special
Author Response
Dear Reviewer 3,
Thank you for valuable comments and recommendations.
Please, find enclosed the answers.
Best regards,
Authors.

Round 2
Reviewer 1 Report
I express my satisfaction that the authors have fully complied with the above recommendations for improving the manuscript, which have been carried out in all sections of the text.
I have no objections to the scientific value and presentation of the article. In general, I have little objection to the style of the English language.
It is common to use passive voice when writing research articles. Unfortunately, the revised text still contains singular wording such as: "we tried (line 52); we found (lines 242-243); we analysed (line 273); We analysed (line 313); we found (line 420); In conclusion, we would like to point out ... (line 493); We have shown (line 505)" and so on. I recommend that these phrases be corrected by observing the passive voice, for example: it was found, it was determined; it was established, etc.
line 306 - the word "im-pact" has a hyphen, please correct this technical error!
Author Response
Dear Reveiwer 1,
Thank you for valuable suggestions and comments. The changes that we made on the manuscript, based on your suggestions were marked on the manuscript by highlighting in yellow.
It is common to use passive voice when writing research articles. Unfortunately, the revised text still contains singular wording such as: "we tried (line 52); we found (lines 242-243); we analysed (line 273); We analysed (line 313); we found (line 420); In conclusion, we would like to point out ... (line 493); We have shown (line 505)" and so on. I recommend that these phrases be corrected by observing the passive voice, for example: it was found, it was determined; it was established, etc.
Reply:
The phrases were corrected into the passive voice.
line 306 - the word "im-pact" has a hyphen, please correct this technical error!
Reply:
The technical error was corrected.
Best regards,
Authors.
Reviewer 2 Report
The issues listed below need to be addressed further.
First, more time and effort should be devoted to revising the manuscript to achieve a more organized structure and accurate and concise presentation.
As the author declared, they have recalculated significant differences using one-way ANOVA followed by Duncan's multiple comparisons (which means mean values of every two groups should be compared). Thus, if there is a significant difference between groups, then all four mean values should be labeled with significance. However, it seems that some mean values in the same row/line have significance markers but some not (e.g. data rows 2(C10:0), 5(C15:0), 12(ΣSFA), 30(ΣUFA), 37(ΣFAshort (from С4 to С10)), 39(ΣFAlong (>С17)) and 40(ΣС4-С16/ΣС17-С24) in Table 2, data lines 3 (1290-1310) and 5(1720-1760)). Correct the significance label and the corresponding result description accordingly. I recommend the authors to carefully verify the significance flags in all tables and figures.
Significance markers are suggested to be added to Fig. S1 where possible. And then combine the revised Figure S1 with Figure 3 or remove the revised Figure S1 (I don’t think the presence of figure S1 in supplementary materials is of much value.).
As to the detailed description of the survey information material covered in Figure 1 (survey methodology/questionnaire/data collected, etc.), the explanation provided is not convincing.
Also, in lines 164-171, the authors should have confirmed the following question. Whether samples of the same tissue were obtained from the same side of the carcass and from a fixed location.
Author Response
We would like to thank Reviewer 2 for valuable suggestions and comments. We present below our reply to each of them separately.
The changes that we made on the manuscript, based on Reviewer 2 suggestions were marked on the manuscript by highlighting in green.
The issues listed below need to be addressed further.
First, more time and effort should be devoted to revising the manuscript to achieve a more organized structure and accurate and concise presentation.
Reply:
Thank you very much for your comment! We’ll pay more efforts to revising our next manuscripts.
As the author declared, they have recalculated significant differences using one-way ANOVA followed by Duncan's multiple comparisons (which means mean values of every two groups should be compared). Thus, if there is a significant difference between groups, then all four mean values should be labeled with significance. However, it seems that some mean values in the same row/line have significance markers but some not (e.g. data rows 2(C10:0), 5(C15:0), 12(ΣSFA), 30(ΣUFA), 37(ΣFAshort (from С4 to С10)), 39(ΣFAlong (>С17)) and 40(ΣС4-С16/ΣС17-С24) in Table 2, data lines 3 (1290-1310) and 5(1720-1760)). Correct the significance label and the corresponding result description accordingly. I recommend the authors to carefully verify the significance flags in all tables and figures.
Reply:
If the mean was not labeled by letter, these mean have no statistical difference from other means. We mentioned in the end of all figures and tables, that “Different letters indicate statistically significant differences (p≤0.05)”.
Significance markers are suggested to be added to Fig. S1 where possible. And then combine the revised Figure S1 with Figure 3 or remove the revised Figure S1 (I don’t think the presence of figure S1 in supplementary materials is of much value.).
Reply:
Thank you for this suggestion! We’ve removed Figure S1 from Supplementary materials.
As to the detailed description of the survey information material covered in Figure 1 (survey methodology/questionnaire/data collected, etc.), the explanation provided is not convincing.
Reply:
We’ve detailed the description in Material and Methods Section.
Following questions were asked: i) how often do you buy pork minced meat ii) how often do your buy chilled pork, frozen pork et cetera over the past 1-2 years?
The consumers had to answer by choosing options- I buy less now/ or rarely, I buy about the same amount (nothing changed), I start buying more often, I never buy pork minced meat/ I never buy pork.
Then the respondents were asked to answer the questions: (1) Why do you buy pork/minced pork, (2) Why you do not buy pork/minced pork? - by choosing the options – affordable price/high price, dietary diversity/reasonability (only in warm period or for holidays for a special grilled meat cocking), satisfied/ not satisfied with quality (fat content), easy/hard to cook, good/ not good sensory characteristics (taste, appearance, juiciness).
Also, in lines 164-171, the authors should have confirmed the following question. Whether samples of the same tissue were obtained from the same side of the carcass and from a fixed location.
Reply:
Yes, the samples of the same tissue were obtained from the same side of the carcass and from a fixed location, as it was described. We also have added the following text in the sentence “….of the carcass and fixed location….”.
Best regards,
Authors.
Reviewer 3 Report
All my questions have been well solved and could be accepted in this form
Author Response
Dear Reviewer 3,
Thank you very much.
Best regards,
Authors.